# Research on Low-Cost Attitude Estimation for MINS/Dual-Antenna GNSS Integrated Navigation Method

**DOI:** 10.3390/mi10060362

**Published:** 2019-05-30

**Authors:** Hailu Wang, Ning Liu, Zhong Su, Qing Li

**Affiliations:** 1University of Beijing Information Science & Technology Beijing Key Laboratory of High Dynamic Navigation Technology, Beijing 100101, China; whl41017@163.com (H.W.); sz@bistu.edu.cn (Z.S.); liqing@bistu.edu.cn (Q.L.); 2School of Automation, Beijing Institute of Technology, Beijing 100084, China

**Keywords:** Global Navigation Satellite System/Micro-electromechanical Systems-Inertial Navigation System (GNSS/MINS) integration, Low-cost, dual-antenna, Extended Kalman Filtering (EKF), attitude

## Abstract

A high-precision navigation system is required for an unmanned vehicle, and the high-precision sensor is expensive. A low-cost, high-precision, dual-antenna Global Navigation Satellite System/Micro-electromechanical Systems-Inertial Navigation System (GNSS/MINS) combination method is proposed. The GNSS with dual antennas provides velocity, position, and attitude angle information as the measurement information is combined with the MINS. By increasing the heading angle, pitch angle, velocity, the accuracy of the integrated system is improved. The Extended Kalman Filtering (EKF) integrated algorithm simulation is designed to verify the feasibility and is realized based on the Field Programmable Gate Array and Advanced RISC Machine (ARM+FPGA) system. Static and dynamic tests were performed using the Synchronous Position, Attitude and Navigation (SPAN-CPT) as a reference system. The results show that the velocity, position, and attitude angle accuracy were improved. The yaw angle and pitch angle accuracy were 0.2° Root Mean Square (RMS) and 0.3° RMS, respectively. The method can be used as a navigation system for the unmanned vehicle.

## 1. Introduction

At present, autonomous vehicles and cooperative intelligent transportation systems (C-ITS) have become the focus of attention [1,2]. Navigation technology is the central concern. At present, high-precision navigation systems are mainly ensured by using expensive sensors, so it is particularly important to develop a low-cost, high-precision, vehicle-integrated navigation system. 

Integrated navigation based on information fusion and prediction algorithms is more advantageous than individual navigation systems [3,4]. Although the Global Navigation Satellite System (GNSS) has higher positioning accuracy than other positioning methods, it is susceptible to various interferences such as multipath effects which include radar, electromagnetic interference, and signal blocks [5,6], contrary to a single-antenna GNSS/Strap-down integrated navigation system (SINS) [7]. An enhanced method for single-antenna Global Position System (GPS) based attitude determination is proposed [8]. The results show that single-antenna/Inertial Navigation System (INS) can estimate the velocity to calibrate the heading of the Inertial Measurement Union (IMU) and provides more accurate navigation information than the single-GNSS system alone. However, the heading angle of the single-antenna has a large measurement error. When the single-antenna/INS is stationary, the system fails to provide the attitude angle and position. When the vehicle moves linearly, the gyro of the INS does not work, the heading angle gradually diverges, and there is a large error in the heading observation of the GNSS. Multi-baseline GNSS can determine attitude by using carrier-phase differential technology [5,9,10]. However, multi-antenna GNSS has a high-cost and complicated structure. Also, the update rate is too low to meet the high bandwidth requirement for the control system to catch the high dynamic attitude change of the vehicle [11]. Multi-array antennas must receive four GNSSs at the same time, so the system becomes fragile and difficult to install and configure for land vehicles. 

In order to improve the accuracy of integrated navigation, researchers began to explore different algorithms to linearize the system [3,12]. The estimation algorithm based on integrated navigation has been widely investigated in recent years. Wei Wang and Zongyu Liu’s work was focused on the combination of GPS and INS navigation research, using the Kalman Filter (KF) and the Extended Kalman Filter (EKF) to compensate the error generated by the information fusion [13]. A decision tree-based multiple-model Unscented Kalman Filter (UKF) for attitude estimation using low-cost MEMS magnetic sensor arrays was proposed [14]. According to the theory provided, the effects of EKF and Unscented Kalman Filter (UKF) are almost the same [15]; however, UKF has a large number of calculations and a complicated algorithm. Cooperative parallel particle filters for online model selection and applications to urban mobility were proposed [16]. They cooperate for providing a global estimator of the variable of interest at the same time. Particle filtering uses some discrete random sampling points to approximate the probability density function of random variables and has higher precision [17,18]. However, the particle filter algorithm is more complicated to realize for embedded devices and has a time delay [19].

Least Squares Support Vector Machine (LS-SVM) has been proposed. Xiyuan Chen and Yuan Xu used the algorithm of LS-SVM on the navigation for indoor mobile robots and achieved good results [20]. In the process of achieving a solution, the operation time and memory space increased rapidly with the dimension of the state matrix. Thus, this method was not suitable for vehicle navigation. The mathematical model of navigation by Abdel–Hafez provides the Adaptive Neural Fuzzy Inference System (ANFIS) algorithm [21], but a large amount of computation and a long learning time creates system delay, which is not ideal for real-time systems for autonomous vehicles. Some researchers also used different combination forms of GPS and INS, such as tightly-coupled GPS/INS navigation [4,13,22,23], which is hard to be realized and too complex to be applied in most industrial products.

With the development of artificial intelligence, Jung using radial basis function neural networks (RBFNN) and dynamic neural networks for training GNSS [24,25], could improve navigation accuracy. Martino proposed a synergistic parallel particle filter. Jian proposed that the uncertainty of the Global Navigation Satellite System/Micro-electromechanical Systems-Inertial Navigation System (GNSS/MINS) dynamic model and the measurement model would reduce the system performance, and an adaptive interactive multi-model filter (IMM) was proposed [26]. Although the above methods have achieved good results in some aspects, and often take a long time to train the parameters of the neural network, they are difficult to implement.

In the project, in addition to considering the type of sensor applied, the environmental conditions of the system application, the complexity of the algorithm, the accuracy, and the cost of hardware equipment should also be considered. The EKF is a good combination filter. Based on the above considerations, we propose a combination of dual-antenna GNSS and a Micro-electromechanical Systems-Inertial Navigation System (MINS). The dual-antenna GNSS can provide accurate velocity, position, heading angle, and pitch angle [27]. A gyroscope can provide a high precision attitude angle in a short time but suffers from accumulating errors caused by the gyro biases. Two-antenna GNSS can provide a very accurate yaw angle and pitch in a long time, but the update rate is low. Thus, it is reasonable for it to be combined with the high update rate derived from gyros. In this way, system navigation performance could improve. EKF is a very practical nonlinear filtering algorithm. Because of its simple structure, it is widely used in the field of integrated navigation [28]. Considering the accuracy and real-time of the algorithm, EKF is used as the combined navigation filtering algorithm. Therefore, the integration of these sensors and filtering algorithms seems to be optimal.

The rest of this paper is organized as follows: Section 2 is the introduction of the reference coordinate frame. Section 3 is the error model of the MEMS-gyro, the description of the calibration parameters, and the solution process of the MINS velocity, position, and attitude. The error propagation model is established for the combined system. Section 4 is the design and analysis of the combined filtering method. In the fifth section, the embedded implementation of the algorithm and the driving test, analysis of the data is provided, and finally a summary of the paper.

## 2. Reference Frame

In physics, an inertial coordinate frame is one that does not accelerate or rotate with respect to the rest of the Universe. This does not define a unique coordinate frame. In navigation, a more specific form of the inertial frame, known as the Earth-centered inertial frame, is used, denoted by the symbol i, as in Figure 1a. Earth-centered Earth-fixed frame, commonly abbreviated to earth frame, as in Figure 1b, is similar to the Earth Centered Inertial (ECI) frame, except that all axes remain fixed with respect to the Earth. The Earth-centered Earth-fixed (ECEF) frame is denoted by the symbol E and has its origin at the center of the ellipsoid modeling the Earth’s surface, which is roughly at the center of mass. 

The local navigation frame, local level navigation frame, geodetic, or geographic frame is denoted by the symbol n, as presented in Figure 2a. Its origin is the point a navigation solution is sought for the navigation system, the user, or the host vehicle’s center of mass. Height and geodetic latitude of a body, sometimes known as the ellipsoidal height, is the distance from a body to the ellipsoid surface along the normal to that ellipsoid, as shown in Figure 2b. 

## 3. System Model Analysis

### 3.1. IMU Model

The IMU model is established as the follow:(1)$$\{\begin{array}{c} \tilde{\omega }=b_{g}+b_{gT}(T)+(S_{g}+M_{g})\omega +w_{g}\\ \tilde{f}=b_{a}+b_{gT}(T)+(S_{a}+M_{a})f+w_{a} \end{array}$$

The model of the gyro is:(2)$$\left[\begin{array}{c} {\tilde{\omega }}_{x}\\ {\tilde{\omega }}_{y}\\ {\tilde{\omega }}_{z} \end{array}\right]=\left[\begin{array}{c} b_{g,x}\\ b_{g,y}\\ b_{g,z} \end{array}\right]+\left[\begin{array}{ccc} s_{g,x} & m_{g,xy} & m_{g,xz}\\ m_{g,yx} & s_{g,y} & m_{g,yz}\\ m_{g,zx} & m_{g,zy} & s_{g,z} \end{array}\right]\left[\begin{array}{c} \omega_{x}\\ \omega_{y}\\ \omega_{z} \end{array}\right]+\left[\begin{array}{c} \omega_{g,x}\\ \omega_{g,x}\\ \omega_{g,x} \end{array}\right]$$

The model of the acceleration is:(3)$$\left[\begin{array}{c} {\tilde{f}}_{x}\\ {\tilde{f}}_{y}\\ {\tilde{f}}_{z} \end{array}\right]=\left[\begin{array}{c} b_{a,x}{+b}_{aT,x}(T)\\ b_{a,y}{+b}_{aT,y}(T)\\ b_{a,z}{+b}_{aT,z}(T) \end{array}\right]+\left[\begin{array}{ccc} s_{a,x} & m_{a,xy} & m_{a,xz}\\ m_{a,yx} & s_{a,y} & m_{a,yz}\\ m_{a,zx} & m_{a,zy} & s_{a,z} \end{array}\right]\left[\begin{array}{c} f_{x}\\ f_{y}\\ f_{z} \end{array}\right]+\left[\begin{array}{c} w_{g,x}\\ w_{g,x}\\ w_{g,x} \end{array}\right]$$

In Equations (2) and (3), $\tilde{\omega }$ and $\tilde{f}$ are the output values of the gyroscope and the accelerometer, respectively. $b_{g}$ and $b_{a}$ are the constant zero bias of the gyro and the acceleration, respectively. $b_{gT}(T)$ and $b_{aT}(T)$ are the zero temperature-related items of the gyroscope and the accelerometer respectively. $S_{g}$ and $S_{a}$ are the calibration factor matrix of the gyroscope and accelerometer, respectively. $M_{a}$ and $M_{g}$ are the cross coupling error coefficient matrices, respectively. $w_{g}$ and $w_{a}$ are the output noise, respectively.

The parameters to be calibrated in the IMU model are the constant zero offset of the gyro, scale factor, cross-coupling error coefficient, zero-bias temperature correlation coefficient, constant zero offset of the acceleration, zero-bias temperature correlation term, scale factor, and cross-coupling term error factor.

#### 3.1.1. Attitude Update

The attitude update step of the ECEF navigation equation uses the angular-rate measurement, $\omega_{ib}^{b}$ to update the attitude solution. This is necessary because the North East axis moves with the rotation of the Earth, and the time derivative of the coordinate transformation matrix is:(4)$${\dot{C}}_{b}^{n}=C_{b}^{n}\Omega_{nb}^{b}$$

This will be divided into three items as follows:(5)$${\dot{C}}_{b}^{n}=C_{b}^{n}\Omega_{ib}^{b}-(\Omega_{ie}^{b}+\omega_{en}^{n})C_{b}^{n}$$

The earth rotation vector in the local navigation coordinate system is given, so the anti-symmetric matrix is:
(6)$$\Omega_{ie}^{n}=\omega_{ie}\left(\begin{array}{ccc} 0 & sinL_{b} & 0\\ -sinL_{b} & 0 & -cosL_{b}\\ 0 & cosL_{b} & 0 \end{array}\right)$$

Notice that this is a function that is only related to the latitude.
(7)$$\Omega_{en}^{n}=\left(\begin{array}{ccc} 0 & -\omega_{en,z}^{n} & \omega_{en,y}^{n}\\ \omega_{en,z}^{n} & 0 & -\omega_{en,x}^{n}\\ -\omega_{en,y}^{n} & \omega_{en,x}^{n} & 0 \end{array}\right)$$

The conversion matrix of the ECEF reference coordinate system to the navigation coordinate system has been given, and the derivative of the time is as follows:(8)$$\omega_{en}^{n}=\left(\begin{array}{c} v_{eb,E}^{n}/(R_{E}(L_{b})+h_{b})\\ -v_{eb,N}^{n}/(R_{N}(L_{b})+h_{b})\\ -v_{eb,E}^{n}tanL_{b}/(R_{E}(L_{b})+h_{b}) \end{array}\right)$$

Obtaining a complete analysis solution is complex, and taking into account changes in position and velocity over the pose update interval may require recursive navigation equations. However, by ignoring this change and truncating the power series expansion of the exponential term to the first order, a reasonable approximation of most applications can be obtained, giving:(9)$$C_{b}^{n}(+)=C_{b}^{n}(-)(I_{3}-\Omega_{ib}^{b}t_{i})-(\Omega_{ie}^{n}(-)+C_{en}^{n}(-))C_{b}^{n}(-)\tau_{i}$$

Here $\Omega_{ie}^{n}(-)$ is calculated by the $L_{b}(-)$, $\Omega_{en}^{n}(-)$ is calculated by $L_{b}(-)$, $h_{b}(-)$, and $v_{eb}^{n}(-)$.

#### 3.1.2. Velocity Update

The transformation of the velocity from ECEF to the local navigation frame is:(10)$$v_{eb}^{n}=C_{e}^{n}v_{eb}^{e}$$

Derived from the above formula:(11)$${\dot{v}}_{eb}^{n}={\dot{C}}_{e}^{n}v_{eb}^{e}+C_{e}^{n}{\dot{v}}_{eb}^{e}$$

Therefore, there is an additional acceleration conversion, centrifugal acceleration, and Coriolis term in the ECEF reference coordinate system. The first term to replace the second term is applied:(12)$${\dot{v}}_{eb}^{n}=-\Omega_{en}^{n}v_{eb}^{n}+C_{e}^{n}(-\Omega_{ie}^{e}\Omega_{ie}^{e}r_{eb}^{e}-2\Omega_{ie}^{e}v_{eb}^{b}+a_{ib}^{e})$$

For specific force accelerations, gravity and centrifugal force are as follows:(13)$${\dot{v}}_{eb}^{n}=f_{ib}^{n}+g_{b}^{n}(L_{b},h_{b})-(\Omega_{en}^{n}+2\Omega_{ie}^{n})v_{eb}^{n}$$

The acceleration due to gravity is modeled as a function of latitude and longitude. So, getting a complete analysis solution is complicated. However, since the Coriolis and transmission rate terms are usually the smallest, it is reasonable to ignore their variation in the integration interval. In general, the variation of the acceleration caused by the integration in the integration interval could usually be ignored.
(14)$$v_{eb}^{n}\approx v_{eb}^{n}(-)+[f_{ib}^{n}+g_{b}^{n}(L_{b}(-),h_{b}(-))-(\Omega_{en}^{n}(-)+2\Omega_{ie}^{n}(-))v_{eb}^{b}(-)]\tau_{i}$$

#### 3.1.3. Position Update

The variation of the meridian and the transverse radii of curvature $R_{N}$ and $R_{E}$ with the geodetic latitude $L_{b}$, is weak, so it is acceptable to neglect their variation with latitude over the integration interval. Assuming that the velocity changes linearly over time within the integration interval, the approximate value of the position update is:(15)$$h_{b}(+){=h}_{b}(-)-\frac{\tau_{i}}{2}(v_{eb,D(-)}^{n}+v_{eb,D(+)}^{n})$$
(16)$$L_{b}(+)=L_{b}(-)+\frac{\tau }{2}(\frac{v_{eb,N(-)}^{n}}{\left(R_{N}(L_{b}(-))+h_{b}(-))cosL_{b}(-\right)}+\frac{v_{eb,E(+)}^{n}}{R_{N}(L_{b}(+))+h_{b}(-))cosL_{b}(+)})$$
(17)$$\lambda_{b}(+)=\lambda_{b}(-)+\frac{\tau_{i}}{2}(\frac{v_{eb,E(-)}^{n}}{\left(R_{E}(L_{b}(-))+h_{b}(-))cosL_{b}(-\right)}+\frac{v_{eb,E(+)}^{n}}{R_{E}(L_{b}(+))+h_{b}(+))cosL_{b}(+)})$$

### 3.2. Schematic of Inertial Navigation Process 

The process of strap-down inertial navigation attitude, velocity, and position update is presented in Figure 3:

### 3.3. System Dynamic Model

This section mainly analyzes the error of the three-dimensional inertial system and dual-antenna GNSS system, using the "mode of error state". In this paper, the study is about position, velocity, and attitude, but the INS estimated indicated a position, velocity, and attitude error. Even in the high- dynamic environment, the velocity, position, and attitude change rapidly, but these error state changes are relatively slow compared with the high frequency dynamic. The slower change property of the estimated state can enhance the stability of the filter and make the overall performance of the filter better. The EKF equation of the state contains 15 system states. The combination model is established for the dual-antenna GNSS and MINS system. 

The INS error propagation matrix is:(18)$${\dot{X}}_{\left(t\right)}=F(t)X_{\left(t\right)}+B_{\left(t\right)}u_{\left(t\right)}+G(t)w(t)$$

Where $\dot{X}(t)$ is the n-dimensional system state vector, F(t) is the *n* × *n* dimensional system dynamic matrix, and u(t) is the control input (control input not modeled in this study; *B* and *u* terms are omitted from the equation).
(19)$$X_{1}={\left[\begin{array}{ccccccccccccccc} \varphi_{E} & \varphi_{N} & \varphi_{U} & \delta_{VE} & \delta_{VN} & \delta_{VU} & \delta L & \delta \lambda & \delta h & \delta f_{x} & \delta f_{y} & \delta f_{z} & \delta_{\omega x} & \delta_{\omega y} & \delta_{\omega z} \end{array}\right]}_{15\times 1}^{T}$$

Includes three attitude errors $\delta_{\alpha }$, $\delta_{\beta }$, $\delta_{\gamma }$, three velocity errors $\delta_{{}_{VN}}$, $\delta_{{}_{VE}}$, $\delta_{{}_{VD}}$, and three position errors δL, δl, δh, x-axis acceleration deviation $\left({m/s}^{2}\right)$, y-axis acceleration deviation $\left({m/s}^{2}\right)$, z-axis acceleration deviation $\left({m/s}^{2}\right)$, x-axis gyro deviation (°/s), y-axis gyro deviation (°/s), and z-axis gyro deviation (°/s).
(20)$$F_{s}={\left[\begin{array}{ccccccccc} 0 & -\Omega sinL & \frac{V_{N}}{R} & 0 & \frac{1}{R} & 0 & -\Omega sinL & 0 & \frac{V_{E}}{R^{2}}\\ F_{21} & 0 & F_{23} & -\frac{1}{R} & 0 & 0 & 0 & 0 & \frac{V_{N}}{R^{2}}\\ -\frac{V_{N}}{R} & F_{32} & 0 & 0 & -\frac{tanL}{R} & 0 & F_{37} & 0 & \frac{V_{E}tanL}{R^{2}}\\ 0 & -f_{D} & f_{E} & \frac{V_{D}}{R} & F_{45} & \frac{V_{N}}{R} & F_{47} & 0 & F_{49}\\ -f_{D} & 0 & f_{N} & F_{54} & F_{55} & F_{56} & F_{57} & 0 & F_{59}\\ -f_{E} & f_{N} & 0 & -\frac{2V_{N}}{R} & F_{65} & 0 & 2\Omega V_{E}sinL & 0 & F_{69}\\ 0 & 0 & 0 & \frac{1}{R} & 0 & 0 & 0 & 0 & -\frac{V_{N}}{R^{2}}\\ 0 & 0 & 0 & 0 & \frac{1}{RcosL} & 0 & \frac{V_{E}tanL}{RcosL} & 0 & -\frac{V_{E}}{R^{2}cosL}\\ 0 & 0 & 0 & 0 & 0 & -1 & 0 & 0 & 0 \end{array}\right]}_{9\times 9}$$

In the formula above:(21)$$F_{21}=\Omega sinL+\frac{V_{E}}{R}tanL$$
(22)$$F_{23}=\Omega cosL+\frac{V_{E}}{R}$$
(23)$$F_{32}=-\Omega cosL-\frac{V_{E}}{R}$$
(24)$$F_{37}=-\Omega cosL-\frac{V_{E}}{R{cos}^{2}L}$$
(25)$$F_{45}=-2(\Omega sinL+\frac{V_{E}}{R}tanL)$$
(26)$$F_{47}=-V_{E}(2\Omega cosL+\frac{V_{E}}{R{cos}^{2}L})$$
(27)$$F_{54}=2\Omega sinL+\frac{V_{E}}{tanL}$$
(28)$$F_{55}=\frac{1}{R}(V_{N}tanL+V_{D})$$
(29)$$F_{56}=2\Omega cosL+\frac{V_{E}}{R}$$
(30)$$F_{57}=2\Omega (V_{N}cosL-V_{D}sinL)+\frac{V_{N}V_{E}}{R{cos}^{2}L}$$
(31)$$F_{57}=2\Omega (V_{N}cosL-V_{D}sinL)+\frac{V_{N}V_{E}}{R{cos}^{2}L}$$
(32)$$F_{65}=-2(\Omega cosL+\frac{V_{E}}{R})$$
(33)$$F_{59}=-\frac{V_{E}}{R^{2}}(V_{N}tanL+V_{D})$$
(34)$$F_{DCM-6\times 6}=\left[\begin{array}{cccccc} -C_{11} & -C_{12} & -C_{13} & 0 & 0 & 0\\ -C_{21} & -C_{22} & -C_{23} & 0 & 0 & 0\\ -C_{31} & -C_{32} & -C_{33} & 0 & 0 & 0\\ 0 & 0 & 0 & C_{11} & C_{12} & C_{13}\\ 0 & 0 & 0 & C_{21} & C_{22} & C_{23}\\ 0 & 0 & 0 & C_{31} & C_{32} & C_{33} \end{array}\right]$$
(35)$$F={\left[\begin{array}{cc} F_{S-9\times 9} & 0_{3\times 6}\\ & F_{DCM-6\times 6}\\ 0_{6\times 15} & \end{array}\right]}_{15\times 15}$$

Where, R is the radius of the earth, $V_{N}$, $V_{E}$, $V_{D}$ is the north, east, and down velocity indicated by inertial navigation, $f_{n}$, $f_{E}$, $f_{D}$ are the northward, eastward, and downward specific force measured by IMU, L is the latitude indicated by inertial navigation, and Ω is the angular velocity of the earth rotation.

## 4. Integrated Navigation Method

### 4.1. Observation

The measurement model is embodied in matrix H, which relates the measurement model to the filtering state. The model selected in this experiment is the attitude angle of the dual-antenna GNSS navigation system, including the yaw angle β and the elevation angle α. The dual-antenna GNSS navigation system cannot provide the roll angle of the vehicle, only the two-dimensional attitude angle. The GNSS navigation system can provide the current position of the vehicle, based on the local geographic coordinate frame as the reference coordinate system, and the longitude $L_{G}$, latitude $l_{G}$, and altitude $h_{G}$ as the navigation parameters for the GNSS measurement information. The frequency of the GNSS navigation is lower than that of the INS. In this experiment, the frequency of GNSS navigation is set to 10 Hz, using the Cartesian vehicle $\left[\begin{array}{ccc} V_{N} & V_{E} & V_{D} \end{array}\right]$ as the measured velocity of the GNSS. 

The position provided by the GNSS receiver can be expressed as the sum of the true value under the geography and the corresponding error, as follows:(36)$$\left[\begin{array}{c} L_{G}\\ l_{G}\\ h_{G} \end{array}\right]=\left[\begin{array}{c} L_{N}-\frac{N_{N}}{R_{N}}\\ l_{N}-\frac{N_{E}}{R_{N}cosL}\\ h_{N}-N_{h} \end{array}\right]$$

The position error of the dual-antenna GNSS relative to the northeast down navigational coordinate system is:(37)$$Z_{P}(t)=\left[\begin{array}{c} (L_{I}-L_{G})R_{M}\\ (l_{I}-l_{G})R_{N}\\ h_{I}-h_{G} \end{array}\right]=\left[\begin{array}{c} R_{M}\delta l+\delta p_{GN}\\ R_{N}\delta l+\delta p_{GE}\\ \delta h+\delta p_{GD} \end{array}\right]=H_{p}X(t)+V_{p}(t)$$

The measurement noise is treated as white noise with variances of $\sigma_{PN}^{2},\sigma_{PE}^{2},\sigma_{PD}^{2}$, which is the product of the pseudo-range measurement error of the GNSS receiver and the corresponding Horizontal Dilution of Precision (HDOP) value. The velocity measurement information of the MINS is expressed as the sum of the true value and the corresponding velocity error, and the velocity measurement information of GNSS is expressed as the true value and the corresponding velocity error as the local navigation framework reference.

The component of the GNSS velocity error in the North East navigation coordinate system. The velocity measurement vector is:(38)$$\begin{array}{ll} Z_{v}(t) & =\left[\begin{array}{c} V_{IN}-V_{GN}\\ V_{IE}-V_{GE}\\ V_{ID}-V_{GD} \end{array}\right]=\left[\begin{array}{c} \delta_{VN}+M_{N}\\ \delta_{VE}+M_{E}\\ \delta_{VD}+M_{D} \end{array}\right]\\ & =H_{V}X(t)+V_{V}(t) \end{array}$$

In the above formula: $H_{V}(t)=\left[0_{3\times 3}\vdots diag\left[\begin{array}{ccc} 1 & 1 & 1 \end{array}\right]\vdots 0_{3\times 12}\right]$.

The measurement noise is the pseudo-range rate $\dot{\rho }$ of the GNSS receiver. The measurement variance is $\sigma_{{}_{\dot{\rho }}}^{2}$, and the standard deviation of the velocity error in the northeast direction is caused by the pseudo-range rate.

The standard deviation of the error is:(39)$$\sigma_{v}=HDOP\times \sigma_{\dot{p}}$$

The attitude measurement vector is:(40)$$Z_{\varphi }(t)=\left[\begin{array}{c} \varphi_{I\alpha }-\varphi_{G\alpha }\\ \varphi_{I\beta }-\varphi_{G\beta } \end{array}\right]=H_{\varphi }X(t)+V_{\varphi }(t)$$

$v_{\varphi }$ is the observation noise.
(41)$$Z=\left[\begin{array}{c} Z_{p(t)}\\ Z_{v(t)}\\ Z_{\varphi (t)} \end{array}\right]={\left[\begin{array}{c} H_{V}\\ H_{P}\\ H_{\varphi } \end{array}\right]}_{8\times 15}\cdot X_{\left(t\right)}+{\left[\begin{array}{c} v_{p}\\ v_{v}\\ v_{\varphi } \end{array}\right]}_{8\times 1}$$

### 4.2. EKF Process and Work Sequence 

In order to construct a discrete EKF filter, the system error state equation is discretized, and the error state of the inertial navigation system is expressed at the moment, indicating the error state of the momentary inertial navigation system:(42)$${\dot{X}}_{k+1}=\Phi_{k+1,k}X_{k}+\Gamma_{k}w_{k}$$

Where $\Phi_{k}$ is $t_{k}$ the system transfer matrix of the moment, which can be expressed by the system matrix *F* shown as follows: (43)$$\Phi_{k}=exp\left[F(t_{t+1}-t_{k})\right]={\sum_{k=0}^{\infty }\left[F(th)T\right]}^{n}/n!$$
(44)$$\Gamma_{k}=\left\{\sum_{n=1}^{\infty }\frac{1}{n!}{\left[F(tk)T\right]}^{n-1}\right\}G(tk)T$$

The nonlinear system state equation with random noise is:(45)$${\dot{X}}_{k+1}=\Phi_{k+1,k}X_{k}+\Gamma_{k}w_{k}$$

Time update is:(46)$$\begin{array}{l} {\hat{x}}_{k|k-1}={\hat{x}}_{k-1}+f({\hat{x}}_{k-1},t_{k-1})T_{s}\\ P_{k|k-1}=\Phi_{k|k-1}P_{k-1}^{}\Phi_{k|k-1}^{T}Q_{k-1} \end{array}$$

Measurement update is:(47)$$\begin{array}{l} {\hat{z}}_{k|k-1}=h({\hat{x}}_{k|k-1},t_{k})\\ K_{k}=P_{k|k-1}H_{k}^{T}{\left[H_{k}P_{k|k-1}H_{k}^{T}+R_{k}\right]}^{-1}\\ {\hat{x}}_{k}={\hat{x}}_{k|k-1}+K_{k}(z_{k}-{\hat{z}}_{k|k-1})\\ P_{k}=[I-K_{k}H_{k}]P_{k|k-1}{\left[I-K_{k}H_{k}\right]}^{T}+K_{k}R_{k}K_{k}^{T} \end{array}$$

$F(t_{k-1})$ is the discrete form of the Jacobian matrix and the Hessian matrix, as follows:(48)$$H_{k}=\frac{\partial h[x(t_{k-1}),t_{k-1}]}{\partial^{T}(t_{k-1})}|\begin{array}{c} x(t_{k-1})={\hat{x}}_{k-1} \end{array}$$
(49)$$\Phi_{k|k-1}\approx I+F(t_{k-1})T_{s}$$

When the EKF process linearizes an approximation of the nonlinear function, it ignores the higher-order terms above the second-order, which is suitable for the weak nonlinear system whose update time interval is approximately linear.

### 4.3. Integrated Navigation Structure

The position, velocity, and attitude angles are used as GNSS measurement information for the EKF to estimate the INS error. The GNSS and INS can work at the same time independently. Such a simple structure can be used for any INS and GNSS equipment, and the filter is not contaminated by MIMU errors, presented in Figure 4.

### 4.4. Simulation Experiment

The EKF algorithm is verified based on MATLAB 2018. The convergence characteristics of EKF can be seen from the curves as shown in Figure 5. The convergence time and accuracy need to be adjusted with the actual equipment and data. The feasibility of the algorithm can be verified by simulation, and the parameters of the EKF design need to be confirmed by simulation.

The convergence characteristics of EKF for the velocity are shown in Figure 5a. The red curve, blue curve, and black curves are the north, east, and down velocity, respectively, as seen in Figure 5a. The velocity covariance converges around 0.2. The blue curve, red curve and black curves are the north, east, and down attitude angle covariance of EKF, respectively, as shown in Figure 5b.

The performance of the EKF in terms of the position is shown in Figure 6. From Figure 6a, the red curve, blue curve, and the pink curve are the covariance of the east position, down position, and the north position, respectively. The red curve is the down position error after the EKF in Figure 6b. The down position error is stabilized at 0.6, while the original attitude error is decreasing.

The performance of the EKF in terms of the attitude angle is shown in Figure 7. The red curve is the original error of the yaw angle in Figure 7a. Due to the error accumulation of the z-axis gyro, the yaw error deviates from the initial condition. The theoretical yaw error at 0.0 is perfect, but which is impossible. The velocity of the deviation for the original error can be adjusted by the MEMS-gyro. As time goes on, the yaw error of the EKF determined coincides with the original yaw error within the margin of the error, and this process is an important characteristic regarding the performance for the EKF.

The performance of convergence of EKF can be determined by simulation. Although MINS provides 100 Hz attitude update rate, the update rate of EKF is only 10 Hz to predict and estimate. If data interruption provided by GNSS is in the 10 Hz frequency interval, the prediction is executed alone, and the complete prediction and update are not performed. This means that the filter estimation error is not consistent with the state of the INS, and the filter estimation error is valid at the actual update time, although it is used for the next cycle of filtering. In the computer simulation, the update rate of the GNSS can be set to meet the dynamic condition requirements. We think this is reasonable, because the error state changes relatively slowly, so it is still very accurate in the update of 10 Hz frequency. This way of processing is exchanged for faster processing time with minimum precision.

## 5. Experiment

### 5.1. Design of the Prototype 

The hardware composition of the MEMS-INS/GNSS integrated navigation system is shown in Figure 8a. Its main components consist of three parts: three single-axis MEMS gyros, three single-axis MEMS accelerometers, AD modules, dual GNSS RF signal receiving antenna, dual GNSS receiver board, and a navigation computer with ARM+FPGA architecture. ARM uses the STM32F767 series (STMicroelectronics manufacturer, Modena and Toulouse, Italy and France) with a 32-bit Cortex-M7 CPU, DPFPU ART accelerator and 2 Mbytes Flash memory (Analog Devices, Inc, ADI manufacture, Norwood, MA, USA) integrated into two banks allowing read-while-write.

The gyro uses the ADXRS646-EP single-axis gyro produced (Analog Devices, Inc, ADI manufacturer, Norwood, MA, USA). It is a high-precision angular velocity sensor with 0.01°/s acceleration random-walk and high-frequency vibration resistance. The measurement range is ±450°/s. The accelerometer is based on the ADXL354 from Analog Devices (ADI manufacturer, Norwood, MA, USA), with a range of ±2 g or ±4 g.

The FPGA adopts A3P600-2FG-144IY high-performance chip produced by Actel manufacturer (New York, NY, USA) This processing chip has the featuring high speed, high bandwidth, and high capacity. It is suitable for high-speed data communication and high-velocity data acquisition. This core board uses two pieces of MT41J256M16HA-125 DDR3 chip (MICRON company, Milpitas, CA, USA), each DDR has a capacity of 4 Gb. Two DDR chips are combined into a 32-bit data bus width, and the read or write data bandwidth between FPGA and DDR3 is as high as possible 25 Gb. The configuration can meet the needs of high-bandwidth data processing.

The receiving board of the high-precision dual-antenna receiving board has the functions of receiving of the GNSS, BDS, GLONASS, and SBAS GNSS system, equipped with the two RS232 external interfaces (MAXIM Integrated Products, San Jose, CA, USA), and offer time service, navigation location. The board can offer the heading angle and pitch angle. The mobile computer is used for data acquisition with an Intel I5-7600 processor, 1.4 GHz dominant frequency, which can collect data for a long time, regardless of the amount of data and the bandwidth of transmission. RS422 (MAXIM Integrated Products, San Jose, CA, USA) to serial communication is connected with the host computer. The serial communication conversion adapter uses the UPort 1150 series (MOXA manufacturer, Taipei, Taiwan), which has the capability of collecting high frequency data and is suitable for the application with high-bandwidth transmission.

### 5.2. Dynamic Calibration of MEMS Devices

Based on the characteristics of the IMU error model of the MEMS gyro affected by temperature changes requires temperature compensation. Using a two-axis turntable with internal temperature control as the calibration device, as shown in Figure 9a. The standard input speed is set to 300°/s for the control turntable. The output speed of the gyroscope after the calibration is shown in the Figure 9b.

### 5.3. Static test

The dual-antenna is installed on the test vehicle and fixed with the bracket. The vehicle’s mid-perpendicular line is parallel. The master-antenna is placed at the front-end. The slave-antenna is placed at the rear end of the vehicle, as shown in Figure 10a. The dual-antenna GNSS/MINS prototype is placed in the middle of the two antennas. The device is fixed to the bracket to reduce the vibration of the device. The vehicle can provide DC-15V power to meet the requirements of the device. The vehicle is parked on the side of the road to start the initial alignment of the dual-antenna /MINS integrated algorithm. The initial position and initial attitude angle of the integrated system will be resolved by the Navigation solution and EKF.

When the vehicle is stationary, the step of the initial alignment starts. The yaw angle converges gradually, and the heading angle and pitch angle are provided by the dual-antenna GNSS as the initial attitude angles, as shown in Figure 10b.

The calibration method of the mounting angle between the dual-antenna and the IMU is accurate. Before testing the performance of the dual-antenna GNSS/MINS system, the mounting angle of the IMU and the dual-antenna need to be calibrated and compensated.

### 5.4. Dynamic Experiment

The driving vehicle test begins after static alignment. The test includes the performance of the MEMS-gyro navigation, the accuracy of the dual-antenna GNSS navigation attitude angle, the yaw angle accuracy of the integrated system, the performance of the EKF, and the navigation performance in terms of short-term and long-term precision. When the dual-antenna GNSS signal is obstructed, the heading angle accuracy of the integrated system is affected. The baseline length of the two antennas is 0.6682 m, so the accuracy of the dual-antenna is not relative to the length of the baseline.

Using the Synchronous Position, Attitude and Navigation system (SPAN-CPT) produced by Novatel as a reference system to compare with the dual-antenna/MINS system. SPAN-CPT is an integration INS/GNSS, the KF integrated filter is used, and the update rate is 100 Hz, as shown in Figure 11a. The APAN-CPT is equipped with a ring laser gyroscope, MEMS accelerometer, which is more accurate than the MEMS gyro. The ring laser gyroscope was tested and found to have a drift of about 0.1°/h. The overall performance is tactical and maintains accuracy for a few minutes. The performance of the heading angle reference system can provide true approximate data to test heading angle accuracy of the two-antenna GNSS/MINS integrated system. The installation error of the device will affect the convergence effect and accuracy of the EKF. There are errors in determining the center position of the GNSS antenna, which will not be considered here and will be further explained in the future.

The test site is the road around the laboratory and the vehicle is as shown in Figure 11b. The point from the start of the vehicle to the end of the trip is a closed loop. Three experiments were analyzed as a group. The attitude angle error at different velocities was tested, as well as the heading angle accuracy of the straight lines and curves. The time and convergence accuracy of the EKF was at test. The initial vehicle is in the stationary, and the initial alignment is performed. The heading angle converged by the EKF gradually, and the heading angle and the pitch angle are provided by the GNSS as the initial attitude angle. 

The blue line represents the yaw provided by the dual-MINS/GNSS integration system and the red curve represents the yaw test provided by SPAN-CPT as the reference system, as shown in Figure 12a. The heading angle of the initial position is the same as the heading angle of the terminal position. Because the dynamics of the GNSS is less, the position of the curve and the change of the tracking yaw angle are delayed. The initial yaw angle is −5.667 degrees based on the geographic north direction. The clockwise direction is the positive angle and the counterclockwise direction is the negative angle. When the yaw angle exceeds the limits of 180°, it will be shown as the opposite direction. So, the range of the yaw is ±180°. The driving test includes whether the heading angle error will accumulate where the testing road is a closed loop path. The mark is made at the initial position and the vehicle returns to the initial coordinate position in a closed loop. The terminal heading angle is −6.443 degrees. The deviation between the initial yaw angle and the terminal yaw angle provided by the SPAN-CPT is used as the operational error. The difference between the initial heading angle and the heading angle is the systematic error. The deviation of the initial yaw angle and the terminal yaw angle from the dual-antenna/MINS is the driving measurement error. The difference of the driving measurement error with the system error is the error of the EKF integrated algorithm. 

In the curved road process, the angle of the curved road is about 90°. The heading angle of the SPAN-CPT is the reference system. The yaw angle error of the curved road test is as shown in Figure 13. The heading angle error is larger than the straight line test. The yaw accuracy of EKF is decreasing gradually and the yaw starts to converge after five seconds, the maximum error is 1.5°, and then converges to 0. The angular velocity is about 9°/s. It can be seen from the curve that the deviation between the heading angle of GNSS/MINS and SPAN-CPT is <0.25° in the initial moment. This interval is about 10 sections and the vehicle is moving straightly. Then the curve road test is ongoing and the dynamics of the vehicle is increasing. The heading angle deviation has large fluctuations from the blue curve. The filtering precision of the EKF is reduced. 

When the GNSS navigation system is in the high-rise building on both sides of the highway or through the overpass, the GNSS signal l will be lost. The number of GNSS received is less, the weak signal leads to the lower signal-to-noise ratio (SNR) of the GNSS, and the large measurement error will exit. The GNSS system will the fail. The attitude angle, velocity and position error provided by the GNSS become larger, as shown in Figure 14a.

When the standard deviation (STD) of the heading angle is between 0.0 and 5.0, there is no obstruction between the GNSS receiver and the satellites, as in Figure 14a. The number of GNSS captured meets the requirements of location. In that case, the attitude angle accuracy is the highest. When the STD is 5.0–90.0, the GNSS signal received by the GNSS receiver can be used in some environment, but the navigation information has a large measurement error. 90.0–180.0 indicates that the signal between the GNSS and the receiver is obstructed completely, and GNSS is unavailable. When the GNSS signal is interrupted, EKF starts estimating the optimal velocity, position, and attitude by the MINS. The heading angle in not valid until the signal is recovered gradually, as shown in Figure 14b.

In the driving test, the pitch angle of the two-antenna GNSS/MINS is tested. Compared with the SPAN-CPT system, the dual-GNSS/MINS combination has better accuracy and dynamics than the MINS and dual-antenna GNSS in the driving vehicle test, as shown in Figure 15.

The blue curve represents the true GNSS position curve, as shown in Figure 16a. The discontinuous points represent the GNSS information as invalid. If the vehicle moves faster in the 10 Hz output frequency or the dual-antenna GNSS is obscured, it will be shown as discontinuous points. The position of the dual-antenna GNSS/MINS combination is displayed as the northeast reference frame, and the trajectory of the travel is marked by the red curve, as shown in Figure 16b. The specific terrain features can be viewed in the actual driving vehicle map, as seen in Figure 17a. 

The dynamic noise matrix EKF algorithm needs to be combined with the specific MEMS gyroscope sensors and the GNSS receiver. There it takes a long time to select the parameters in the process debugging of the EKF. The dynamic noise matrix Q of the system in the experiment is shown in Table 1.

These system noise values affect the error that is assigned to each state at the time of the prediction. They even affect the filtering covariance seriously, which means they affect the confidence of each state in the filter estimation. The driving test results of the dual-antenna GNSS/MINS are compared with the SPAN-CPT reference system, and the statistical results of the position, velocity, and attitude angle accuracy are shown in Table 2.

The position of the integrated system is imported into Google Earth with longitude, latitude, and altitude as navigation parameters. Google Earth adopts the World Geodetic System-1984 Coordinate System (WGS-84) reference coordinate system. The WGS-84 is the north east ground reference coordinate system. 

The actual track of the driving test, as shown in Figure 17a. In the area, the two-antenna GNSS/MINS estimates the position parameters optimally during the dual-antenna GNSS navigation as invalid. The trajectory of the driving test is enlarged as shown in Figure 17b.

## 6. Conclusions

This paper proposes a navigation method of the dual-antenna GNSS when integrated with the MINS by the EKF integrated algorithm. The SPAN-CPT is used to verify the static and dynamic precision as a reference system. The results show that the accuracy of the attitude angle is similar to the SPAN-CPT system. In the process of a short loss of the GNSS signal, the heading angle is still valid, which helps to improve the stability of the system. The parameters of the system noise need to be adjusted to meet the dynamics of the vehicle. In this experiment, only the SPAN-CPT system was compared. In the future, different reference systems will be compared to reflect the performance differences, such as the single-GNSS/MINS combination system. A filter with a small amount of calculation and high filtering accuracy will be studied and designed.

## Figures and Tables

**Figure 1 micromachines-10-00362-f001:**
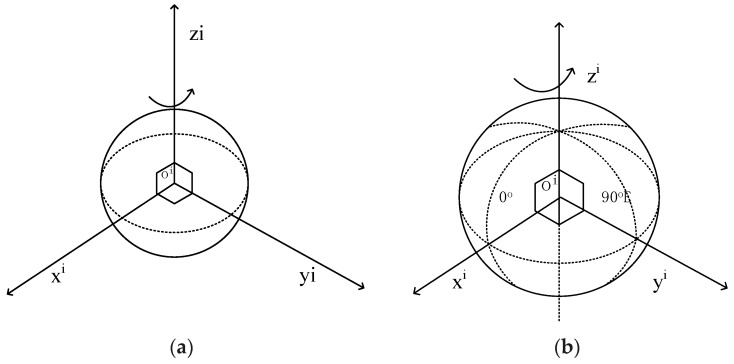
(**a**) Axes of the Earth-Centered Inertial Frame. This is nominally centered at the Earth’s center of mass and oriented with respect to the Earth’s spin axis and the stars. The rotation shown is that of the Earth with respect to space. The z-axis always points along the Earth’s axis of rotation from the center to the north pole (true, not magnetic); (**b**) Axes of the Earth-Centered Earth-fixed Frame. The z-axis always points along the Earth’s axis of rotation from the center to the North Pole (true, not magnetic). The x-axis points from the center to the intersection of the equator with the reference meridian (IRM) or conventional zero meridian (CZM), which defines 0° longitude.

**Figure 2 micromachines-10-00362-f002:**
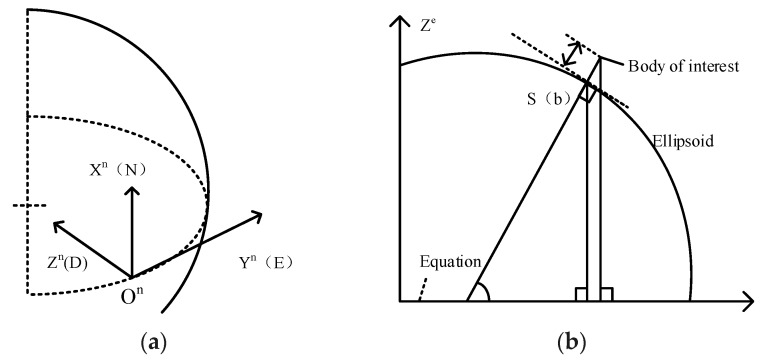
(**a**) Axes of the Local Navigation Frame. The z axis, also known as the down (D) axis, is defined as the normal to the surface of the reference ellipsoid, pointing toward the center of the Earth roughly. True gravity deviates from this slightly due to local anomalies. The x-axis, or north (N) axis, is the projection in the plane orthogonal to the z-axis of the line from the user to the north pole. By completing the orthogonal set, the y-axis always points east and is hence known as the east (E) axis; (**b**) Height and geodetic latitude of a body.

**Figure 3 micromachines-10-00362-f003:**
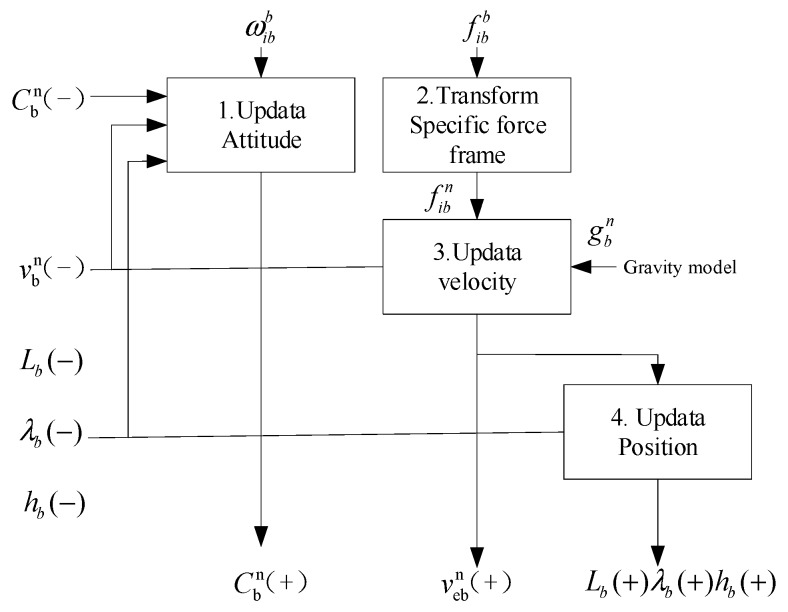
Block diagram of local-navigation-frame equation. (+) represents the *t* and (−) represents the (*t* − *τ*_0_).

**Figure 4 micromachines-10-00362-f004:**
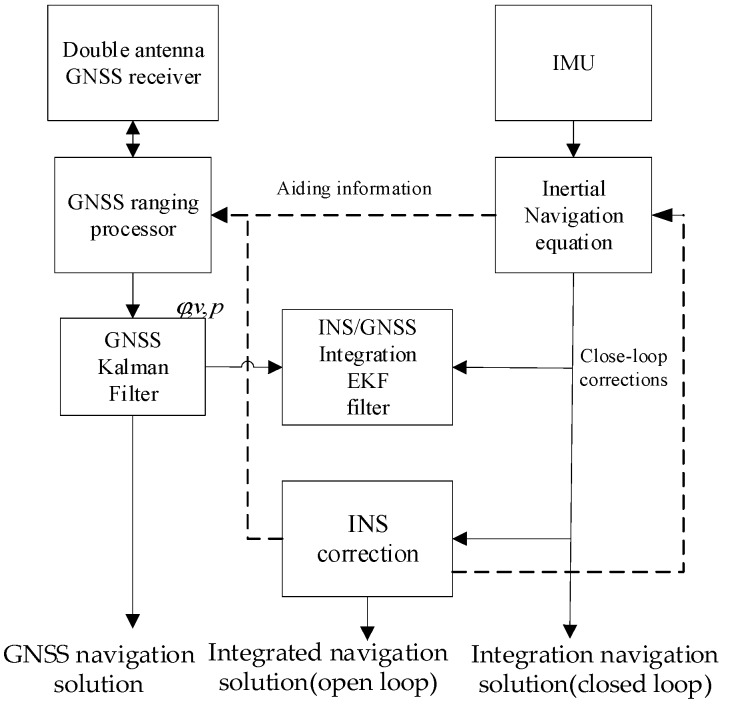
The integrated navigation architecture.

**Figure 5 micromachines-10-00362-f005:**
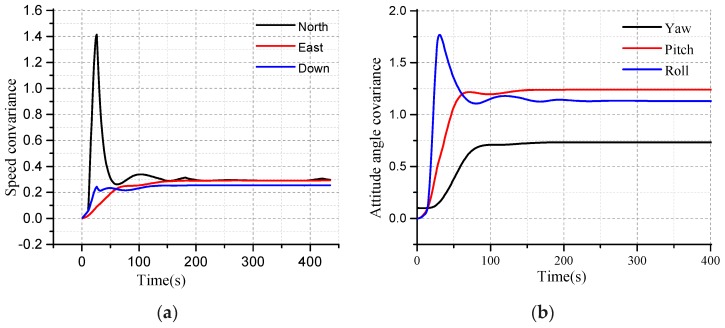
(**a**) The convergence characteristics of the velocity covariance; (**b**) The convergence characteristics of the attitude angle for the Extended Kalman Filtering (EKF).

**Figure 6 micromachines-10-00362-f006:**
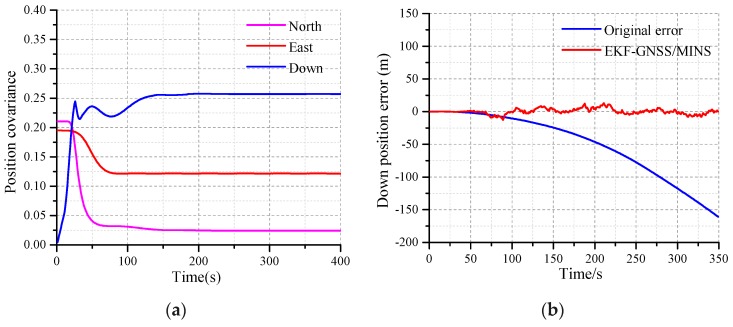
(**a**) Covariance of the east, north, and down positions; (**b**) The characteristic of the up position.

**Figure 7 micromachines-10-00362-f007:**
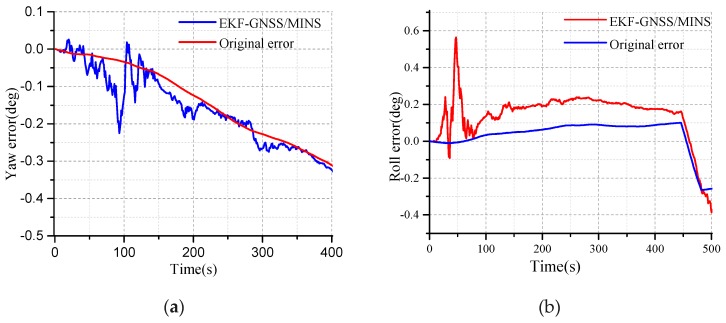
(**a**) The convergence characteristic of the yaw error; (**b**) The convergence characteristics of the roll angle error.

**Figure 8 micromachines-10-00362-f008:**
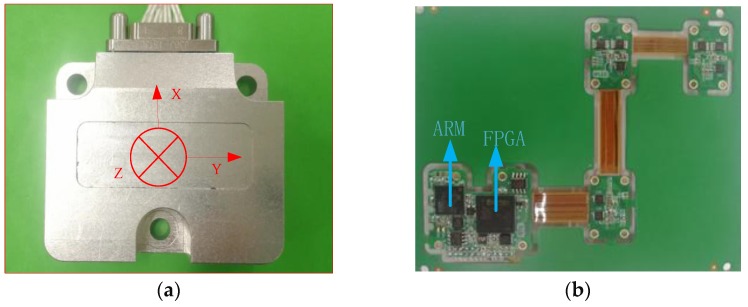
(**a**) The device appearance of Micro-electromechanical Systems-Inertial Navigation System (MINS); (**b**) The integrated circuit system.

**Figure 9 micromachines-10-00362-f009:**
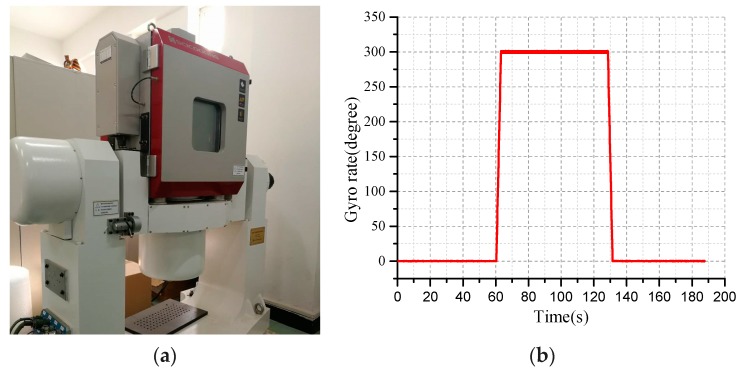
(**a**) The two-dimensional temperature control turntable; (**b**) The output of the gyroscope after calibration.

**Figure 10 micromachines-10-00362-f010:**
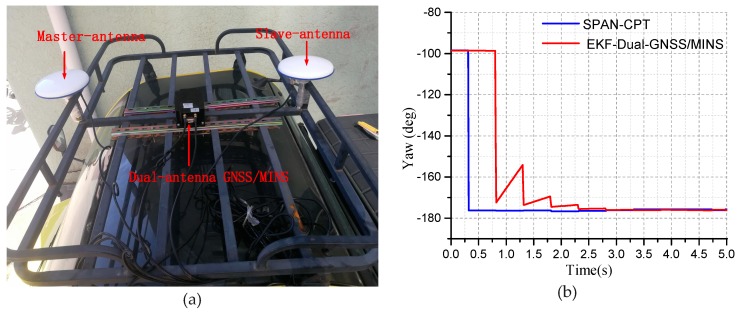
(**a**) The installation instruction of the dual-antenna/MINS; (**b**) The convergence of the yaw for the static test.

**Figure 11 micromachines-10-00362-f011:**
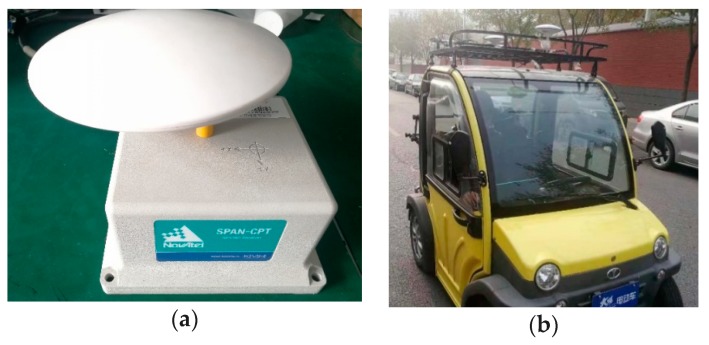
(**a**) The Synchronous Position, Attitude and Navigation (SPAN-CPT) reference system; (**b**) The driving vehicle.

**Figure 12 micromachines-10-00362-f012:**
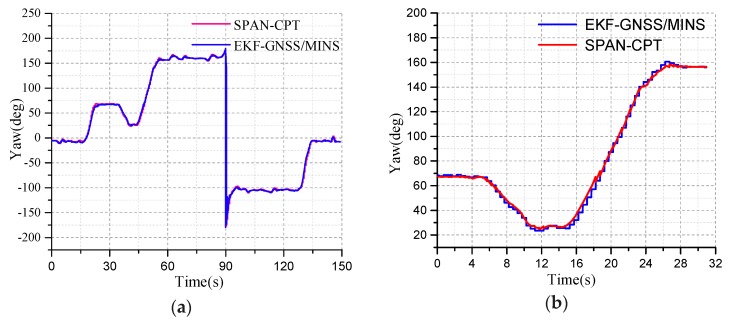
(**a**) The yaw angle test of whole road; (**b**) The yaw angle test of the curved road.

**Figure 13 micromachines-10-00362-f013:**
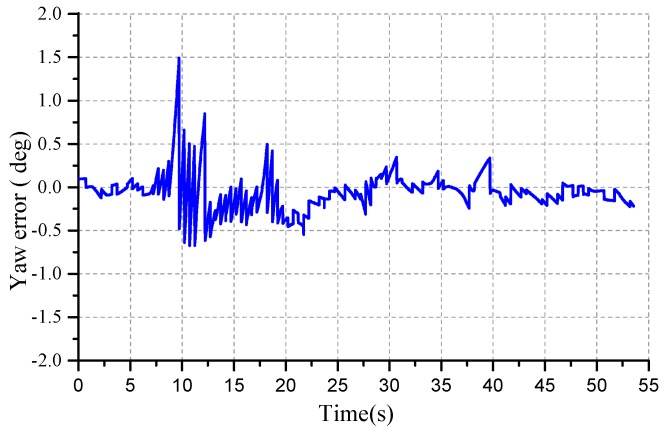
The yaw error of the curved road test.

**Figure 14 micromachines-10-00362-f014:**
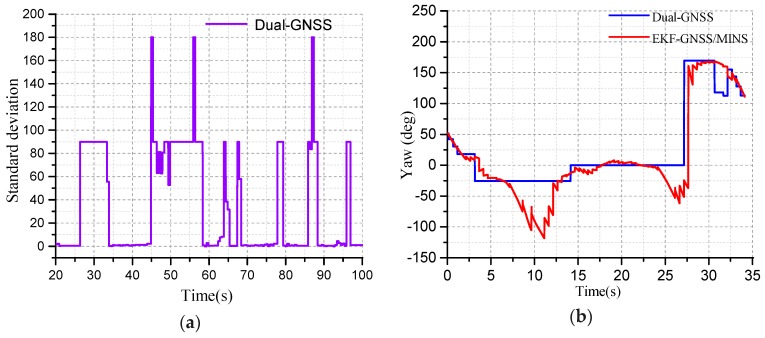
(**a**) The standard deviation of GNSS heading angle; (**b**) The yaw angle test of the dual-antenna GNSS is unavailable.

**Figure 15 micromachines-10-00362-f015:**
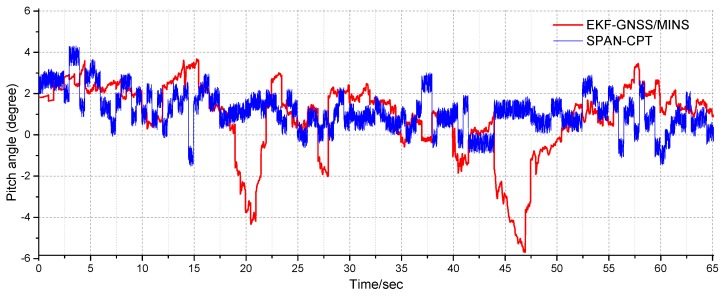
The pitch angle test of the driving test.

**Figure 16 micromachines-10-00362-f016:**
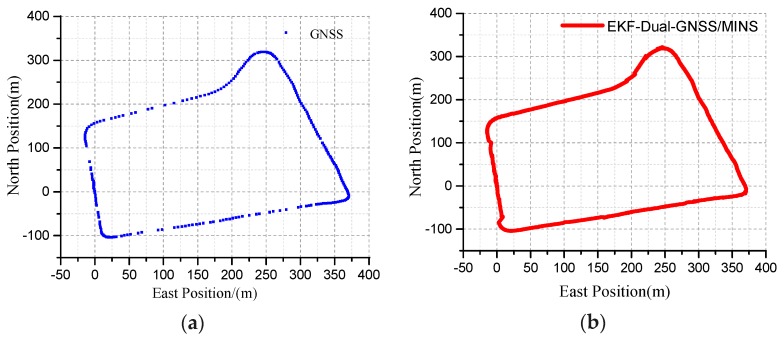
(**a**) The position of the dual-antenna GNSS is blocked; (**b**)The position of the dual-antenna GNSS/MINS integration.

**Figure 17 micromachines-10-00362-f017:**
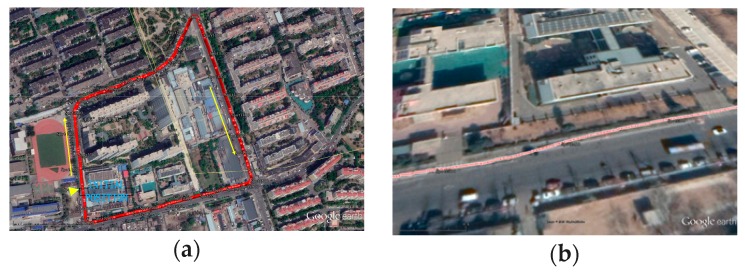
(**a**) The top view of the driving test trajectory from Google Earth; (**b**) The magnified map of the test location.

**Table 1 micromachines-10-00362-t001:** System Dynamic Noise.

Filter State	The Noise Value	Filter State	The Noise Value	Filter State	The Noise Value
$\delta_{\alpha }$	$(3\times 10^{-3}{)}^{2}s$	$\delta_{\beta }$	$(3\times 10^{-3}{)}^{2}s$	$\delta_{\lambda }$	$(3\times 10^{-3}{)}^{2}s$
$\delta_{VE}$	$(1\times 10^{-3}m/s{)}^{2}s$	$\delta_{VN}$	$(1\times 10^{-3}m/s{)}^{2}s$	$\delta_{VU}$	$(1\times 10^{-3}m/s{)}^{2}s$
δL	$(6\times 10^{-9}m{)}^{2}s$	δλ	$(6\times 10^{-9}m{)}^{2}s$	δh	$(6\times 10^{-9}m{)}^{2}s$
δf	$(9\times 10^{-4}{m/s}^{2}{)}^{2}s$	$\delta f_{y}$	$(9\times 10^{-4}{m/s}^{2}{)}^{2}s$	$\delta f_{z}$	$(9\times 10^{-4}m/s^{2}{)}^{2}s$
$\delta_{\omega x}$	$(0.01{}^{\circ}/s{)}^{2}s$	$\delta_{\omega y}$	$(0.01{}^{\circ}/s{)}^{2}s$	$\delta_{\omega z}$	$(0.01{}^{\circ}/s{)}^{2}s$

**Table 2 micromachines-10-00362-t002:** The statistical table of the driving test.

Performance Parameter	Axial	Parameter Value (°)
Position Error RMSE (m)	North	0.1542
	East	0.1442
	Down	0.2212
Velocity Error RMSE (m)	North	0.2321
	East	0.2451
	Down	0.3264
Attitude Error RMSE (°)	Pitch	0.3342
	Yaw	0.2514

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
