# Peer review of "Research on Low-Cost Attitude Estimation for MINS/Dual-Antenna GNSS Integrated Navigation Method"

_micromachines, 2019, doi:10.3390/mi10060362_

Round 1

Reviewer 1 Report

This manuscript describes the attitude estimation using MINS and Dual Antenna GNSS system. The authors provide the A-to-Z background knowledge on the coordinate and system model in detail, and the results of field data is deserve to be evaluated very positively. Overall structure of the paper is very solid and well balanced. However, I could not find any contribution of this paper. Dual-/multi GNSS antenna attitude estimation has been already suggested 10 years before, and IMU/GNSS integration is one of the traditional topics in the navigation field. The authors need to make their contribution more clear and all the field test should be the evidences to prove their contribution as well as to show overall performances of their products.

The detailed comments are;

All the abbreviations need to be explained at the first appearance; SOTM, EKF, UKF, RBFNN, MAST, and so on.

English proof reading is definitely needed

the attitude Angle can converge (p2, L59)

Attitude is free drift based on (p2, L84)

the rest of the Universe.as Figure 1 (p3, L95)

… and so on.

Notation of each term in equations should be defined.

There is no definition of (-) and (+) in equations from (18) t9 (21),  φE, φN, φU in equation (23)

I cannot find δα, δβ, δγ from the equation (23), and the terms from δfx to δwz  should be also matched to each definition.

What do the subscripts I and G in equations from 40 to 44? And there are no definitions of φ, φ, φ,, and φ,  in equation 44

The authors described how they integrated IMU and GNSS in details, but the knowledge described from chapter 2 to 4.2 is not necessary to be written for this manuscript because any one can get it from a number of textbooks. The authors should focus on their contributions and what is different from the others. For example, how they can get the heading information from multi-array GNSS antenna(see C.Kee, et al, Efficient Attitude Determination Using GPS Multiple antennas; Geometrical Concept, and B. Park, et al, A closed-form method for the attitude determination using GNSS Doppler measurements) or what is different when heading information is integrated to the IMU from a general integration of IMU/GNSS.

In a similar way, all the field test results should be the evidences to prove that its suggestions improve the performance of the IMU/GNSS integration, therefore they should prepare their results with that of existing IMU/GNSS, not with that of GNSS-only.

Literature survey

- Attitude based on the single antenna GNSS has been suggested from several papers such as S. Park, 'Enhanced method for single-antenna GPS-based attitude determination’ published before the authors’ cited papers.

Author Response

To Micromachines:

Dear Editor(s) / Reviewer,

Thank you for your letter and for the reviewer’s comments concerning our manuscript entitled " Research on Low-cost attitude Estimation for MINS/Dual-Antenna GNSS Integrated Navigation Method". Those comments are all valuable and very helpful for revising and improving our paper, as well as the important guiding significance to our researches. We have studied comments carefully and have made correction which we hope meet with approval. The main corrections in the paper are as flowing:

1.Revised English usage and grammatical errors.

2.Revised the introduction

3.Revised the reference.

4.Further explanation of the test results.

5.Add test results

6.Made changes and explanations to the title of the diagram

7.Compare with the filtering methods in other papers

8.Each term in equations are defined

9.Remove common formulas

10.Compare with other angle combination methods.

The responds to the reviewer’s comments are as flowing:

Reviewer # 1:

1.Comment:However, I could not find any contribution of this paper.

Response:  We have further explained the main content of this article in the revised manuscript. Dual-antenna GNSS provides speed and position, two-dimensional attitude angles including heading and pitch angles, combined with MINS, MINS can output speed, position and three-dimensional attitude angles, low-cost MEMS device noise and error, through The antenna GNSS combination improves the accuracy of the attitude angle, speed and position.

2.Comment:Dual-/multi GNSS antenna attitude estimation has been already suggested 10 years before and IMU/GNSS integration is one of the traditional topics in the navigation field.

Response: Single-antenna and multi-antenna attitude measurement methods have been proposed. There are still some problems to be solved. Satellites are susceptible to signal interference and occlusion. They are unstable in urban environments and have low update rates. They need to be combined with other angle sensors to obtain Stable navigation performance. To meet the requirements of environment, navigation accuracy, system stability and dynamics during driving for the vehicle.

3. Comment: The authors need to make their contribution more clear and all the field test should be the evidences to prove their contribution as well as to show overall performances of their products”

Response: Thank you for your comments, Further explanation of the contents and results of this paper is presented. A detailed description of the data processing and conclusions is made by comparing the high-precision SPAN-CPT system of NovAtel to reflect the performance of the dual-antenna GNSS/MINS device and the filtering accuracy and convergence time of the extended-man filter algorithm.

4.Comment: All the abbreviations need to be explained at the first appearance; SOTM, EKF, UKF, RBFNN, MAST, and so on.”.

Response: The abbreviations have been explained.

5.Comment: English proof reading is definitely needed”.

Response: Thank you for your comments .The English have been proofed. We have found a teacher whose native language is English, and made systematic changes to the grammar of the paper.

6.Comment: Notation of each term in equations should be defined.

Response: The each term in equation have defined.

7.Comment: “The authors described how they integrated IMU and GNSS in details, but the knowledge described from chapter 2 to 4.2 is not necessary to be written for this manuscript because any one can get it from a number of textbooks”.

Response: Thanks for your comment, we have simplified the Chapter 2-4.2, which will be omitted from the commonly used formula, and the formula for rotating the reference coordinate frame will be omitted. We have further explained the key formula of this article and analyzed it in the last part of the introduction.

8.Comment: “The authors should focus on their contributions and what is different from the others, For example, how they can get the heading information from multi-array GNSS antenna or what is different when heading information is integrated to the IMU from a general integration of IMU/GNSS.”

Response: Thank you for your comments, we are very touched. Multi-array antennas are not suitable for installation on the top of vehicles due to their slow update rate due to their complex installation and configuration. Multi-array antennas require four satellite signals per antenna to resolve the attitude angle, but the system becomes fragile due to signal occlusion.

9.Comment: In a similar way, all the field test results should be the evidences to prove that its suggestions improve the performance of the IMU/GNSS integration, therefore they should prepare their results with that of existing IMU/GNSS, not with that of GNSS-only”

Response: Thank you for your comments. The comparison was made with the SPAN-CPT produced by Nov-Atel. SPAN-CPT is a tightly coupled integrated navigation device with integrated GPS and INS. The IMU is a fiber optic gyroscope (FOG) that can be equipped with a single antenna or dual antenna. Pitch angle accuracy is 0.05 RMS and yaw angle is 0.01 RMS

10.Comment: Attitude based on the single antenna GNSS has been suggested from several papers such as S. Park, 'Enhanced method for single-antenna GPS-based attitude determination’ published before the authors’ cited papers.”

Response: We are very grateful for the problem you mentioned.  In the paper you mentioned . The single-antenna attitude method as a back-up or fault diagnostic system more useful is more useful. The visibility of GPS satellites affects the accuracy of the aircraft's velocity that is the main source of single-antenna attitude. In addition, to use the aircraft dynamic model is natural because single-antenna attitude is for exclusive use of aircraft. We are studying navigation in mobile vehicles, with some differences compared to spacecraft, such as speed and environmental conditions.

We tried our best to improve the manuscript and made some changes in the manuscript. These changes will not influence the content and framework of the paper. And here we did not list the changes but marked with “Track Changes”.

We appreciate for Editors/Reviewer’s warm work earnestly, and hope that the correction will meet with approval.

Once again, thank you very much for your comments and suggestions.

Sincerely Yours,

Ning Liu

Reviewer 2 Report

In my opinion, this work contains interesting idea. However, it could be improved. See my suggestions below.

- First of all, the explanation of your contribution in the introduction is very short (just few lines) before section 2. Please, improve the description of your contribution.

-  Please, improve the caption of Figures 1, 2, 3 and 4 (giving more explanation). 

- Can you idea applied also using a particle filter instead of a EKF? Please, discuss and see my point  below.

- It is important to improve the state-of-the-art discussing other filtering techniques (as particle filtering) to increase the impact and the appeal of the paper.  I suggest  to discuss it and to add some important, relevant and recent references, such as:

F. Gustafsson, G. Hendeby, "Some Relations Between Extended and Unscented Kalman Filters”, IEEE Transactions on  Signal Processing, , vol.60, no.2, pp.545-555, 2012.

P. M. Djuric, J.H. Kotecha, J. Zhang, Yufei Huang,  T. Ghirmai, M.F. Bugallo, J. Miguez, "Particle filtering," in IEEE Signal Processing Magazine, vol. 20, no. 5, pp. 19-38, Sept. 2003.

 L. Martino, J. Read, V. Elvira, F. Louzada, Cooperative Parallel Particle Filters for on-Line Model Selection and Applications to Urban Mobility, Digital Signal Processing Vol. 60, pp. 172-185, 2017.

L. Martino, V. Elvira, G. Camps-Valls, "Group Importance Sampling for Particle Filtering and MCMC", Digital Signal Processing Volume 82, Pages 133-151, 2018

 It can increase the number of interested readers of your work (increasing your audience).

- Please, upload the final version of your manuscript in Arxiv and/or ResearchGate when/if published, to increase the diffusion  and the possible citations of your work.

Author Response

Dear Editor(s) / Reviewer,

Thank you for your letter and for the reviewer’s comments concerning our manuscript entitled " Research on Low-cost attitude Estimation for MINS/Dual-Antenna GNSS Integrated Navigation Method". Those comments are all valuable and very helpful for revising and improving our paper, as well as the important guiding significance to our researches. We have studied comments carefully and have made correction which we hope meet with approval. The main corrections in the paper are as flowing:

1.Revised English usage and grammatical errors.

2.Revised the introduction

3.Revised the reference.

4.Further explanation of the test results.

5.Add test results

6.Made changes and explanations to the title of the diagram

7.Compare with the filtering methods in other papers

8.Each term in equations are defined

9.Remove common formulas

10.Compare with other angle combination methods.

The responds to the reviewer’s comments are as flowing:

Reviewer # 2:

1. Comment: “The explanation of your contribution in the introduction is very short (just few lines) before section 2.”
Response: Explain the contents and results of this paper in the introduction

2. Comment: “Please, improve the caption of Figures 1, 2, 3 and 4 (giv-ing more explanation).”
Response: We have improved the caption of the legend of Figure 1,2,3 and 4,and made further explanation to these Figures.

3. Comment: “Can you idea applied also using a particle filter instead of a EKF? Please, discuss and see my point below“.

Response: We have also considered particle filtering. Considering that the accuracy of particle filtering is higher than that of EKF. But the algorithm more complicated. The existing hardware system we designed does not support the implementation of such an algorithm. We will design the hardware structure and discuss it in the next study.

4. Comment: “It is important to improve the state-of-the-art discussing other filtering techniques (as particle filtering) to increase the impact and the appeal of the paper. I suggest to discuss it and to add some important, relevant and recent references”
Response: Thank you for your suggestion, we have discussed the particle filtering and other state-of-the-art filtering techniques. And added some relevant and re-cent references in the revised manuscript.

We tried our best to improve the manuscript and made some changes in the manuscript. These changes will not influence the content and framework of the paper. And here we did not list the changes but marked with “Track Changes”.

We appreciate for Editors/Reviewer’s warm work earnestly, and hope that the correction will meet with approval.

Once again, thank you very much for your comments and suggestions.

Sincerely Yours,

Ning Liu

Round 2

Reviewer 1 Report

Thanks for the author's sincere reply.

Most of the previous comments are properly reflected in the revised manuscript, but the final results does not yet show the author's contribution clearly.Velocities estimated from a single GNSS antenna can calibrate the IMU heading and a conventional GNSS-MINS integration also improve GNSS position availability. Thus the author should compare their dual-GNSS/MINS results with single GNSS/INS method, not with GNSS.

The authors still need an English proof reading and academic reviewing. There still are several typos such as; 

- line 57

- fn in line 225

- Figure 3, 5,6,7 and others are not cited in the manuscript

- numbering of figures : jump between figure 7 and 10, 11 then 10

- as Figure 16? Figure 16 has not appeared yet.

Author Response

To Micromachines:

Dear Editor(s) / Reviewer,

Thank you for your letter and for the reviewer’s comments concerning our manuscript entitled " Research on Low-cost attitude Estimation for MINS/Dual-Antenna GNSS Integrated Navigation Method". Those comments are all valuable and very helpful for revising and improving our paper, as well as the important guiding significance to our researches. We have studied comments carefully and have made correction which we hope meet with approval. The main corrections in the paper are as flowing:

1.Revised English usage and grammatical errors.

2.Revised the introduction.

3.Revised the sequence of all the figures.

4.Revised the illustration of the test result.

5.Revised the English proof and reading.

6.Academic modification for the manuscript.

7.Revised illustration of the method proposed.

We tried our best to improve the manuscript and made some changes in the manuscript. These changes will not influence the content and framework of the paper. And here we did not list the changes but marked with “Track Changes”.

We appreciate for Editors/Reviewer’s warm work earnestly, and hope that the correction will meet with approval.

Once again, thank you very much for your comments and suggestions.

Sincerely Yours,

Ning Liu

Reviewer 2 Report

The paper has been substantially improved. The paper is now ready for publication.

Author Response

To Micromachines:

Dear Editor(s) / Reviewer,

Thank you for your letter and for the reviewer’s comments concerning our manuscript entitled " Research on Low-cost attitude Estimation for MINS/Dual-Antenna GNSS Integrated Navigation Method". Those comments are all valuable and very helpful for revising and improving our paper, as well as the important guiding significance to our researches. We have studied comments carefully and have made correction which we hope meet with approval. The main corrections in the paper are as flowing:

1.Revised English usage and grammatical errors.

2.Revised the introduction.

3.Revised the sequence of all the figures.

4.Revised the illustration of the test result.

5.Revised the English proof and reading.

6.Academic modification for the manuscript.

7.Revised illustration of the method proposed.

The responds to the reviewer’s comments are as flowing:

Reviewer # 2:

1.Response to comment: “English language and style are fine/minor spell check required”

Response: English language and style have been checked.

We tried our best to improve the manuscript and made some changes in the manuscript. These changes will not influence the content and framework of the paper. And here we did not list the changes but marked with “Track Changes”.

We appreciate for Editors/Reviewer’s warm work earnestly, and hope that the correction will meet with approval.

Once again, thank you very much for your comments and suggestions.

Sincerely Yours,

Ning Liu
